# An Improved Online Self-Calibration Method Utilizing Angular Velocity Observation for Ultra High Accuracy PIGA-Based IMU

**DOI:** 10.3390/s22218136

**Published:** 2022-10-24

**Authors:** Yongfeng Zhang, Shuling Hu, Gongliu Yang, Xiaojun Zhou, Hongwu Liu

**Affiliations:** 1School of Instrumentation and Optoelectronic Engineering, Beihang University, Beijing 100083, China; 2Beijing Institute of Aerospace Control Devices, Beijing 100083, China

**Keywords:** inertial measurement unit (IMU) calibration, pendulous integrating gyroscopic accelerometer (PIGA), angular observation

## Abstract

In the field of ultra high accuracy inertial measurement unit (IMU), pendulous integrating gyroscopic accelerometer (PIGA) has become a research hot spot due to its high-end performance. However, PIGA is sensitive to angular velocity, and the calibration process of PIGA-based IMU will be very complicated, which makes online self-calibration difficult to implement. To solve the above problems, we proposed an online self-calibration method utilizing angular velocity observation. The main contributions of this study are twofold: (1) An error analysis of PIGA is conducted in this paper, and the error model has also been simplified to suit the self-calibration model. (2) An improved online self-calibration method utilizing angular observation based on a simplified PIGA error model is proposed in this study. Experimental results show that the self-calibration method proposed in this study can improve the PIGA online calibration accuracy effectively (with the accuracy within 0.02 m/s/pulse), which can improve the dynamic accuracy of the PIGA.

## 1. Introduction

In the past few decades, pendulous integrating gyroscopic accelerometer (PIGA) has become a research hot spot due to its high-end performance in the field of ultra high accuracy inertial measurement unit (IMU) [1]. The accuracy of gyroscopes has made great progress in the past few decades [2]. However, the accuracy of the accelerometer has a very high correlation between the horizontal attitude and the positioning accuracy of the inertial navigation system (INS), especially in free inertial model [3]. PIGA has many advantages, such as large overload, high accuracy, and high measure range [4]. After a complicated calibration process, PIGA-based IMU can reach high navigation performance [5], which lays the foundation for its utilization in ultra high accuracy INS.

However, PIGA is sensitive to angular velocity, and the calibration process of PIGA-based IMU will be very complicated, which makes online self-calibration difficult to implement. Many researchers have studied the self-calibration method for rotation INS (RINS) and hybrid INS (HINS). In terms of biases estimation of IMU, Fong et al. proposed methods for in-field calibration of IMU without external equipment [6]. In Ref. [7], a gyro-bias calibration method has been proposed for analytic coarse alignment. Han et al. proposed a method for bias calibration of ring laser gyroscopes (RLGs) in single-axis RINS [8]. Furthermore, Li et al. in [9] analyzed the observability of the IMU bias self-calibration method of single-axis RINS. When the rotational degrees of freedom exceed 1, the calibration of IMU’s scale factors can be realized [10]. PIGA contains the angular velocity coupling calibration parameters, with only one rotation axis (only for bias calibration) that cannot meet the self-calibration demand of PIGA.

Many researchers have studied IMU all parameters (biases, scale factors, nonlinear factors, lever arms, etc.) and self-calibration methods for RINS. Ren et al. proposed a multi-position self-calibration method for dual-axis RINS [10]. Yu et al. added gyro scale factor asymmetry factors into the calibration filtering process to improve the self-calibration accuracy of gyroscopes [11]. In Ref. [12], researchers considered the accelerometers-size-effect in self-calibration method for tri-axis RINS. Wen et al. proposed a 40-dimensional error model for the self-calibration process by considering the gyro-accelerometer asynchronous time in dual-axis RINS [13]. The observation method has been analyzed in [14] for vehicle-based INS. The above calibration methods only consider the error parameters under the condition of linear error, there are some studies that have analyzed the calibration methods in nonlinear conditions. In Ref. [15], a KF-based AdaGrad algorithm is proposed to solve the nonlinear problem. Furthermore, Cai et al. extended the dimension of the Kalman filter (KF) model to 51 for ultra high accuracy dual-axis RINS and adopted the RTS smoothing method to improve the calculation rate [16]. In Ref. [17], Pan et al. considered the accelerometer nonlinear scale factor for calibration methods. In Ref. [18], the gyro bias caused by geomagnetic fields in dual-axis RINS has been analyzed and calibrated. However, few researchers have studied the calibration methods for accelerometers with angular velocity coupling parameters. With limited rotation equipment, self-calibration methods for ultra high accuracy PIGA-based IMU are necessary.

The angular observation method has been researched in [19], which is effective in IMU calibration when the encoders of the turntable are accurate [20]. However, the two methods do not apply to the calibration of the PIGA-based IMU, because the angular rate accuracy and differential accuracy obtained from the encoders are poor. To solve the above problems, an improved online self-calibration method utilizing angular velocity observation for ultra high accuracy PIGA-based IMU is proposed in this study. The main contributions of this study are twofold: (1) An error analysis of PIGA is conducted in this paper, and the error model has also been simplified to suit the self-calibration model. (2) An improved online self-calibration method utilizing angular observation based on a simplified PIGA error model is proposed in this study. With only the angular velocities provided by the gyroscopes embedded in the PIGA-based IMU, the angular velocity coupling factors can be estimated utilizing the proposed 43-dimensional self-calibration filtering model. Experimental tests are carried out to verify the feasibility and applicability of the investigated method.

The remainder of this work is presented as follows: Section 2 gives the error analysis of the PIGA, and a simplified error model of PIGA for self-calibration is proposed. Section 3 derives the proposed online self-calibration method utilizing angular observation based on a simplified PIGA error model, and we derive a 43-dimensional filtering model to solve the aforementioned problems. Experimental setup, results, and discussions are provided in Section 4 to verify the effectiveness of the proposed self-calibration method. Finally, the conclusions are given in Section 5.

## 2. Modeling and Analysis of PIGA

### 2.1. Kinetics Analysis of PIGA

The internal coordinate system of PIGA is shown as Figure 1:

In Figure 1, $\dot{\alpha }$ denotes the angular velocity of the outer frame assembly about the $oy_{1}$ axis, $\dot{\beta }$ is the angular velocity of the inner frame assembly about the ox axis relative to the outer frame. $\dot{\phi }$ denotes the angular velocity of the rotor assembly about its axis of rotation relative to the inner frame.

The establishment process of the PIGA internal coordinate system can be expressed as follows: First, it is assumed that the shell coordinate system $ox_{0}y_{0}z_{0}$ of the gyro accelerometer, the outer frame coordinate system $ox_{1}y_{1}z_{1}$, and the inner frame coordinate system oxyz are completely coincident, and the center of mass of the eccentric mass is located on the *z*-axis. Secondly, it is assumed that the outer ring component of PIGA is rotated around the axis by an angle, the inner ring component is rotated by α angle around the axis, and the non-orthogonal state of the PIGA internal coordinate system is obtained as shown in the above figure.

Under the condition that the rotor is fully dynamically balanced, each axis of the coordinate system $ox_{R}y_{R}z_{R}$ is the main axis of inertia of the rotor, so its inertia product is zero, and the calculation formula of the angular momentum of the rotor is as follows:(1)$$\left[\begin{array}{c} H_{x_{R}}\\ H_{y_{R}}\\ H_{z_{R}} \end{array}\right]=\left[\begin{array}{c} A\left(\omega_{x}-\Delta \phi_{y}\omega_{z}\right)\\ B\omega_{y}\\ C\left(\Delta \phi_{y}\omega_{x}+\omega_{z}+\dot{\phi }\right) \end{array}\right]$$

A=B is the moment of inertia about the two coordinate axes perpendicular to the rotor axis. *C* is the moment of inertia of the rotor shaft. Since the ox, oy, and oz axes are not the main inertial axes of the inner frame assembly, their inertia products are not equal to zero. The formula for calculating the angular momentum of the inner frame assembly (excluding the rotor) relative to the oxyz axis is:(2)$$\left[\begin{array}{c} H_{x}\\ H_{y}\\ H_{z} \end{array}\right]=\left[\begin{array}{ccc} J_{xx} & -J_{xy} & -J_{xz}\\ -J_{xy} & J_{yy} & -J_{yz}\\ -J_{xz} & -J_{yz} & J_{zz} \end{array}\right]\left[\begin{array}{c} \omega_{x}\\ \omega_{y}\\ \omega_{z} \end{array}\right]=\left[\begin{array}{c} J_{xx}\omega_{x}-J_{xy}\omega_{y}-J_{xz}\omega_{z}\\ -J_{xy}\omega_{x}+J_{yy}\omega_{y}-J_{yz}\omega_{z}\\ -J_{xz}\omega_{x}-J_{yz}\omega_{y}+J_{zz}\omega_{z} \end{array}\right]$$
where $J_{xx}$, $J_{yy}$, $J_{zz}$ are the moment of inertia of the frame component to the coordinate system oxyz. $J_{xy}$, $J_{xz}$ and $J_{yz}$ are the inertia product of the inner frame component on the planes to which the *x*-axis and the *y*-axis, the *x*-axis and the *z*-axis, and the *y*-axis and the *z*-axis belong.

$M_{x}$ is the total external torque acting on the inner ring shaft, including elastic torque, friction torque, electromagnetic interference torque, etc., which are not related to specific force, and inertial torque, unequal elastic torque, damping torque, etc., which are related to specific force. $M_{x}$ can be described as:(3)$$M_{x}=M_{x}\left(a\right)+M_{xr}$$
where $M_{xr}$ is the sum of the disturbance moments of the inner frame.

Assuming that the coordinates of the center of mass of the PIGA inner ring component are (0,0,l), the moment generated by the acceleration can be obtained, and $M_{xr}$ can be analyzed and calculated below.
(4)$$M_{x}\left(a\right)=F\times l=\left[\begin{array}{ccc} 0 & -ma_{z} & ma_{y}\\ ma_{z} & 0 & -ma_{x}\\ -ma_{y} & ma_{x} & 0 \end{array}\right]\left[\begin{array}{c} 0\\ 0\\ l \end{array}\right]=\left[\begin{array}{c} mla_{y}\\ -mla_{x}\\ 0 \end{array}\right]$$
where *m* is the mass of the inner frame assembly.

The expressions of the input-specific force of PIGA in the inner and outer ring coordinate systems can be expressed as:(5)$$\begin{array}{c} \left[\begin{array}{c} a_{x}\\ a_{y}\\ a_{z} \end{array}\right]=\left[\begin{array}{ccc} 1 & 0 & 0\\ 0 & \cos\beta & \sin\beta \\ 0 & -\sin\beta & \cos\beta \end{array}\right]\left[\begin{array}{ccc} 1 & \Delta \varphi_{z} & 0\\ -\Delta \varphi_{z} & 1 & 0\\ 0 & 0 & 1 \end{array}\right]\left[\begin{array}{c} a_{x_{1}}\\ a_{y_{1}}\\ a_{z_{1}} \end{array}\right]\\ =\left[\begin{array}{c} a_{x_{1}}+\Delta \varphi_{z}a_{y_{1}}\\ a_{y_{1}}\cos\beta +a_{z_{1}}\sin\beta -\Delta \varphi_{z}a_{x_{1}}\cos\beta \\ -a_{y_{1}}\sin\beta +a_{z_{1}}\cos\beta +\Delta \varphi_{z}a_{x_{1}}\sin\beta \end{array}\right] \end{array}$$

According to the above formula, the inertia moment caused by the input-specific force is:(6)$$\begin{array}{c} M_{x}\left(a\right)=ml\left[a_{y_{0}}\cos\beta +\left(a_{x_{0}}\sin\alpha +a_{z_{0}}\cos\alpha \right)\sin\beta \right.\\ \left.-\Delta \phi_{z}\left(a_{x_{0}}\cos\alpha -a_{z_{0}}\sin\alpha \right)\cos\beta \right] \end{array}$$

### 2.2. Simplified PIGA Error Model

After the above analysis of various error sources, to accurately calibrate the nonlinear error term of the PIGA, it is necessary to accurately model the gyroscopes.

It can be seen from the working principle of PIGA that the motion equation of the inner frame shaft of PIGA is deduced according to the Euler equation $\frac{dH}{dt}+\omega \times H=M$. The precession angle α is much larger than the misalignment angle β, and β is a small angle. Now, let $I_{x}=J_{xx}+A$, $I_{y}=J_{yy}+B$, $I_{z}=J_{zz_{R}}$, $H_{z_{R}}=H$, and simplify processing, Then cosβ=cos2β=1, sinβ=β, sin2β=2β, deriving the output equation of PIGA can obtain:(7)$$\begin{array}{c} \dot{\alpha }=\frac{ml}{H}a_{y_{1}}+\frac{ml}{H}\beta \left(a_{x_{0}}\sin\alpha +a_{z_{0}}\cos\alpha \right)\;-\;\frac{\Delta \phi_{z}}{H}\left(a_{x_{0}}\cos\alpha \;-\;a_{z_{0}}\sin\alpha \right)\\ +\frac{1}{H}\left(M_{x}\left(B\right)\;+\;M_{x_{R}}\right)\;-\;\left(\omega_{y_{0}}\;+\;\omega_{z_{1}}\beta \;-\;\Delta \phi_{z}\omega_{x_{1}}\right)\;-\;\frac{I_{x}\;+\;\Delta \phi_{z}J_{xy}}{H}{\dot{\omega }}_{x_{1}}\\ +\frac{J_{xy}\;-\;I_{x}\Delta \phi_{z}\;-\;\left(A\Delta \phi_{y}\;+\;J_{xz}\right)\beta }{H}\left(\ddot{\alpha }\;+\;{\dot{\omega }}_{y_{0}}\right)\;+\;\frac{A\Delta \phi_{y}\;+\;J_{xz}\;+\;J_{xy}\beta }{H}{\dot{\omega }}_{z_{1}}\\ +\frac{J_{xy}\;+\;\beta J_{xz}\;+\;\Delta \phi_{z}\left(I_{z}\;-\;I_{y}\right)}{H}\omega_{x_{1}}\omega_{z_{1}}\;+\;\frac{J_{xz}\;-\;J_{xy}\beta \;-\;\Delta \phi_{z}\left(I_{z}\;-\;I_{y}\right)}{H}\omega_{x_{1}}\omega_{y_{0}}\\ +\frac{J_{xy}\Delta \phi_{z}\;-\;\left(I_{z}\;-\;I_{y}\right)\;+\;4\beta J_{yz}}{H}\omega_{z_{1}}\omega_{y_{0}}\;+\;\frac{J_{xz}\;-\;J_{xy}\beta \;-\;\Delta \phi_{z}\left(I_{z}\;-\;I_{y}\right)}{H}\omega_{x_{1}}\dot{\alpha }\\ +\frac{J_{xy}\Delta \phi_{z}\;-\;\left(I_{z}\;-\;I_{y}\right)\;+\;4\beta J_{yz}}{H}\omega_{z_{1}}\dot{\alpha }\;+\;\frac{2J_{xz}\Delta \phi_{z}\;+\;2\beta \left(I_{z}\;-\;I_{y}\right)\;+\;2J_{yz}}{H}\omega_{y_{0}}\dot{\alpha }\\ +\frac{J_{xy}\Delta \phi_{z}\;+\;\beta \left(I_{z}\;-\;I_{y}\right)\;+\;J_{yz}}{H}{\dot{\alpha }}^{2}\;-\;\frac{J_{xz}\Delta \phi_{z}}{H}\omega_{x_{1}}^{2}\;+\;\frac{J_{xz}\Delta \phi_{z}\;+\;\beta \left(I_{z}\;-\;I_{y}\right)\;+\;J_{yz}}{H}\omega_{y_{0}}^{2}\\ -\frac{\left(I_{z}\;-\;I_{y}\right)\beta \;+\;J_{yz}}{H}\omega_{z_{1}}^{2} \end{array}$$

Since the output equation of PIGA in the ideal state can be expressed as $\dot{\alpha }=\frac{ml}{H}a_{y_{1}}$, that is, in the above formula, $\dot{\alpha }=\frac{ml}{H}a_{y_{1}}$ is an effective signal, and the rest can be regarded as error terms, and its physical meaning is as follows: $\frac{M_{m}}{H}$ has nothing to do with the input specific force, mainly the error term caused by elastic torque, friction torque and electromagnetic interference torque caused by electromagnetic components. $\omega_{y_{0}}+\omega_{z_{1}}\beta$ is the error term due to the involved motion of the gyro shell. $\Delta \phi_{z}$ and $\Delta \phi_{y}$ are the error terms caused by the PIGA outer ring shaft and the inner ring shaft, as well as the out system between the inner ring shaft and the rotor, which are affected by machining and assembly errors. ${\dot{\alpha }}^{2}$ is the nonlinear error of the gyroscopic accelerometer, which is caused by the centrifugal force of the inner ring assembly as it rotates around the outer ring axis.

Based on Equation (Equation 7), the general error model equation of PIGA can be obtained as follows:(8)$$\begin{array}{c} \dot{\alpha }=K_{0}+K_{x}a_{x}+K_{y}a_{y}+K_{z}a_{z}+K_{xx}a_{x}^{2}+K_{yy}a_{y}^{2}+K_{zz}a_{z}^{2}+K_{xy}a_{x}a_{y}\\ +K_{xz}a_{x}a_{z}+K_{yz}a_{y}a_{z}+{\dot{\Delta }}_{x}\dot{\omega }{}_{x}+{\dot{\Delta }}_{y}\dot{\omega }{}_{y}+{\dot{\Delta }}_{z}\dot{\omega }{}_{z}+\Delta_{x}\omega_{x}+\Delta_{y}\omega_{y}+\Delta_{z}\omega_{z}\\ +\Delta_{xx}\omega_{x}^{2}+\Delta_{yy}\omega_{y}^{2}+\Delta_{zz}\omega_{z}^{2}+\Delta_{xy}\omega_{x}\omega_{y}+\Delta_{xz}\omega_{x}\omega_{z}+\Delta_{yz}\omega_{y}\omega_{z}\\ +\delta_{x}a_{y}\omega_{x}+\delta_{y}a_{y}\omega_{y}+\delta_{z}a_{y}\omega_{z} \end{array}$$
where *K* is the static error coefficient, Δ denotes the dynamic error coefficient, δ is the mixed error coefficient.

The above PIGA error model expression includes the static error model of PIGA, which is the term related to the linear acceleration of PIGA; the dynamic error model, which is the term related to the input angular velocity and angular acceleration; the mixed error model, which is related to the linear acceleration and PIGA.

Since the input shaft and output shaft of PIGA are coincident, the corresponding terms of the cross-axis *x*-axis and *z*-axis have little effect on the output. The simplified model of Equation (Equation 8) can be written as:(9)$$\dot{\alpha }=K_{0}+K_{y}a_{y}+K_{yy}a_{y}^{2}+K_{\omega }\omega_{y}$$

It can be seen from the above analysis that $K_{0}$ is introduced by unequal elastic torque, friction torque, damping torque, and other torques that have nothing to do with the input specific force; $K_{y}$ is caused by the change of the meter parameters, and the change includes the detection quality and the motor rotor quality. Changes, pendulum length changes, changes in the inertia radius of the motor rotor moment of inertia, and changes in the motor speed, among which the change in mass is small and generally ignored; the quadratic term coefficient $K_{yy}$ is proposed due to the existence of ${\dot{\alpha }}^{2}$, and they are determined by the inertia of the inner frame component. Product, unequal inertia and rotational inertia, etc., are caused by the centrifugal force of the inner frame shaft due to the rotation of the gyro accelerometer around the outer frame shaft under the condition of the verticality error; $K_{\omega }$ is caused by the rotation of the output shaft. The error caused by the angular velocity coupling term. Since the coefficient $K_{yy}$ of the gyro accelerometer error model is affected by the interference torque existing on the inner frame axis, when machining the gyro accelerometer, the interference torque existing on the inner frame axis can be utilized to reduce its impact on the scale factors.

## 3. Online Self-Calibration Method Utilizing Angular Velocity Observation

### 3.1. Error Model of PIGAs in Filtering

As the core component of the strategic weapon IMU system, the gyro accelerometer’s calibration level directly affects and determines the actual combat effectiveness of the weapon [18]. In the actual system use, when the rocket engine is turned off during the transition from power flight to ballistic flight, the gyro accelerometer inertial group is required to control the cut-off speed and initial pitch, and heading of the missile. Setup, simulation analysis, etc., consist of only two PIGAs and a quartz accelerometer.

The IMU accelerometer inertial group in this section consists of two gyro accelerometers and a quartz accelerometer (hereinafter referred to as the accelerometer component). The calibration parameters include the accelerometer bias, scale factor, installation error angle, quadratic term coefficient, and angle Rate coupling term coefficients.

Considering that the accelerometers in the IMU are not strictly orthogonally installed, in the inertial navigation solution, the output of each inertial device must be projection is in the same orthogonal coordinate system, which is named the IMU coordinate system (*m*-frame) in this paper.

In this work, to convert the measurement information of the accelerometer from the oblique coordinate system to the IMU coordinate system, an orthogonal coordinate system (*o*-frame) is defined by the sensitive axis of the accelerometer. The $x_{o}$ axis is consistent with the sensitive axis $x_{a}$ of the accelerometer, and the $y_{o}$ axis is in the plane formed by the sensitive axes $x_{a}$ and $y_{a}$ of the accelerometer and differs from the $y_{a}$ axis by a small angle $\beta_{yz}$, the $z_{o}$ axis is the accelerometer sensitive axis $z_{a}$ rotated by a small angle $\beta_{zx}$ around the $x_{a}$ axis, and then rotated by a small angle $\beta_{zy}$ around the $y_{a}$ axis, as shown in Figure 2.

The installation error matrix of the accelerometer can be expressed as:(10)$$M_{a}^{o}\approx \left[\begin{array}{ccc} 1 & 0 & 0\\ -\beta_{yz} & 1 & 0\\ \beta_{zy} & -\beta_{zx} & 1 \end{array}\right]$$
where $\beta_{ij}$ represents the deflection angle of the *i*-axis of the accelerometer sensitive axis around the *j*-axis of the *o*-frame.

Since the installation error between the gyro component and the accelerometer component is a small angle, the coordinate transformation matrix from the system to the IMU coordinate system can be written as:(11)$$C_{o}^{m}\approx \left[\begin{array}{ccc} 1 & -\eta_{z} & \eta_{y}\\ \eta_{z} & 1 & -\eta_{x}\\ -\eta_{y} & \eta_{x} & 1 \end{array}\right]$$
where the installation error angle $\eta_{i}$ between the gyro component and the accelerometer component is the Euler angle concerning the *i*-axis.

The installation error matrix of the accelerometer can be defined by 6 small angles, as shown in Figure 3, and in the IMU coordinate frame:(12)$$M_{a}^{m}=M_{a}^{o}C_{o}^{m}\approx \left[\begin{array}{ccc} 1 & -\eta_{z} & \eta_{y}\\ \eta_{z}-\beta_{yz} & 1 & -\eta_{x}\\ -\eta_{y}+\beta_{zy} & \eta_{x}-\beta_{zx} & 1 \end{array}\right]=\left[\begin{array}{ccc} 1 & \alpha_{xz} & -\alpha_{xy}\\ -\alpha_{yz} & 1 & \alpha_{yx}\\ \alpha_{zy} & -\alpha_{zx} & 1 \end{array}\right]$$

The unit vectors measured by the accelerometer components in the IMU are $ox_{a}$, $oy_{a}$, and $oz_{a}$, respectively, and the order of magnitude of the cubic term of the instrument is very small, so the actual accelerometer inertial group input and output model is simplified as:(13)$$N_{a}=K_{1}^{a}\left(M_{a}^{o}f^{a}+K_{2}^{a}{\left(f^{a}\right)}^{2}+b^{a}+K_{\omega }\omega +v^{a}\right)$$

Neglecting the accelerometer noise term, we can obtain:(14)$$f^{m}\approx K_{A}N_{a}-K_{A2}{\left(f^{m}\right)}^{2}-b^{m}-K_{\omega }\omega$$
where $f^{m}$ is the specific force measured by the accelerometer. $K_{A}=M_{a}^{m}{\left(K_{1}^{a}\right)}^{-1}$. Expand Equation (Equation 14) as: (15)$$\left[\begin{array}{c} f_{x}^{m}\\ f_{y}^{m}\\ f_{z}^{m} \end{array}\right]=\left[\begin{array}{ccc} K_{x}^{a} & \alpha_{xz} & -\alpha_{xy}\\ -\alpha_{yz} & K_{y}^{a} & \alpha_{yx}\\ \alpha_{zy} & -\alpha_{zx} & K_{z}^{a} \end{array}\right]\left[\begin{array}{c} N_{x}^{a}\\ N_{y}^{a}\\ N_{z}^{a} \end{array}\right]-\left[\begin{array}{c} K_{2x}^{a}{\left(N_{x}^{a}\right)}^{2}\\ K_{2y}^{a}{\left(N_{y}^{a}\right)}^{2}\\ K_{2z}^{a}{\left(N_{z}^{a}\right)}^{2} \end{array}\right]-\left[\begin{array}{c} b_{x}^{m}\\ b_{y}^{m}\\ b_{z}^{m} \end{array}\right]-\left[\begin{array}{c} K_{\omega }^{x}\omega_{x}\\ K_{\omega }^{y}\omega_{y}\\ K_{\omega }^{z}\omega_{y} \end{array}\right]$$

### 3.2. 43-Dimensional Kalman Filtering Model for Self-Calibration

In the process of self-calibration, there is a lever arm at the observation point of the velocity position and the rotation center of the IMU, which is called the outer lever arm in this paper. If the rotation center of the turntable does not coincide with the IMU, the observation of the velocity and position will have errors along with the rotation of the indexing mechanism. Therefore, it is necessary to analyze the effect of the outer lever arm.

Assuming that the outer lever arms between the IMU and the turntable are $\delta l_{x}^{b}$, $\delta l_{y}^{b}$ and $\delta l_{z}^{b}$, the velocity and position observations of the IMU can be written as:(16)$$\begin{array}{c} v_{obv}=v_{e}^{n}+C_{b}^{n}\left(\omega_{eb}^{b}\times \delta l^{b}\right)\\ p_{obv}=p+\left[\begin{array}{ccc} \frac{1}{R_{M}\;+\;h} & 0 & 0\\ 0 & \frac{1}{(R_{N}\;+\;h)\cos L} & 0\\ 0 & 0 & 1 \end{array}\right]C_{b}^{n}\delta l^{b} \end{array}$$
where $R_{M}$ and $R_{N}$ are the earth radius parameter, *h* is the height.

It should be noted that in Equation (Equation 15), The non-orthogonal angles of the gyro and accelerometer are usually only a few tens of arc minutes, so the formula can be written as: (17)$$\left[\begin{array}{c} f_{x}^{m}\\ f_{y}^{m}\\ f_{z}^{m} \end{array}\right]\approx \left[\begin{array}{ccc} K_{x}^{a} & \alpha_{xz} & -\alpha_{xy}\\ -\alpha_{yz} & K_{y}^{a} & \alpha_{yx}\\ \alpha_{zy} & -\alpha_{zx} & K_{z}^{a} \end{array}\right]\left[\begin{array}{c} N_{x}^{a}\\ N_{y}^{a}\\ N_{z}^{a} \end{array}\right]-\left[\begin{array}{c} K_{2x}^{a}{\left(N_{x}^{a}\right)}^{2}\\ K_{2y}^{a}{\left(N_{y}^{a}\right)}^{2}\\ K_{2z}^{a}{\left(N_{z}^{a}\right)}^{2} \end{array}\right]-\left[\begin{array}{c} b_{x}^{m}\\ b_{y}^{m}\\ b_{z}^{m} \end{array}\right]-\left[\begin{array}{c} K_{\omega }^{x}N_{x}^{g}\\ K_{\omega }^{y}N_{y}^{g}\\ K_{\omega }^{z}N_{z}^{g} \end{array}\right]$$

In this paper, a 43-dimensional Kalman filter is designed to estimate the error and calibration parameters of the IMU. The state quantities in the error equation include the scale error, zero bias, inner lever arm, outer lever arm, and gyro acceleration of the gyroscope and accelerometer. The state quantity of the filtering method proposed in this paper can be written as:(18)$$X={\left[\begin{array}{cccccccc} \varphi^{T} & \delta v_{e}^{nT} & \delta p^{T} & {X_{g}}^{T} & {X_{a}}^{T} & \delta l^{bT} & \delta r^{bT} & \delta t_{a} \end{array}\right]}^{T}$$
where $\delta p={\left[\begin{array}{ccc} \delta L & \delta \lambda & \delta h \end{array}\right]}^{T}$, $X_{g}$ is the gyroscope calibration error parameters. $X_{a}$ represents the accelerometer calibration error parameters. $\delta l^{bT}$ is the outer lever arm vector, and $\delta r^{bT}$ is the inner lever arm vector. The gyro error vector and accelerometer error vector are shown in Equation (Equation 19).
(19)$$\begin{array}{c} X_{g}={\left[\begin{array}{cccccc} \delta k_{11}^{g} & \delta k_{21}^{g} & \delta k_{31}^{g} & \delta k_{22}^{g} & \delta k_{32}^{g} & \delta k_{33}^{g} \end{array}\begin{array}{ccc} \varepsilon_{x} & \varepsilon_{y} & \varepsilon_{z} \end{array}\right]}^{T}\\ X_{a}=[\begin{array}{cccccc} \delta k_{11}^{a} & \delta k_{21}^{a} & \delta k_{31}^{a} & \delta k_{12}^{a} & \delta k_{22}^{a} & \delta k_{32}^{a} \end{array}\\ \begin{array}{ccccccccc} \delta k_{13}^{a} & \delta k_{23}^{a} & \delta k_{33}^{a} & \nabla_{x} & \nabla_{y} & \nabla_{z} & \delta K_{2x}^{a} & \delta K_{2y}^{a} & \delta K_{2z}^{a} \end{array}\begin{array}{ccc} \delta K{{}_{\omega }^{a}}_{x} & \delta K{{}_{\omega }^{a}}_{x} & \delta K{{}_{\omega }^{a}}_{x} \end{array}{]}^{T} \end{array}$$

The state function of KF can be described as:(20)$$\dot{X}=\mathrm{FX}+\mathrm{Gu}$$

According to the previous error analysis of the IMU, the state transition matrix F can be obtained as:(21)$$F=\left[\begin{array}{cccccccc} -\left[\omega_{in}^{n}\times \right] & F_{12} & F_{13} & F_{14} & 0_{3\times 15} & 0_{3\times 3} & 0_{3\times 3} & 0_{3\times 1}\\ \left[(C_{b}^{n}f^{b})\times \right] & F_{22} & F_{23} & 0_{3\times 9} & F_{25} & 0_{3\times 3} & F_{27} & F_{28}\\ 0_{3\times 3} & F_{32} & F_{33} & 0_{3\times 9} & 0_{3\times 15} & 0_{3\times 3} & 0_{3\times 3} & 0_{3\times 1}\\ 0_{9\times 3} & 0_{9\times 3} & 0_{9\times 3} & 0_{9\times 9} & 0_{9\times 15} & 0_{9\times 3} & 0_{9\times 3} & 0_{9\times 1}\\ 0_{15\times 3} & 0_{15\times 3} & 0_{15\times 3} & 0_{15\times 9} & 0_{15\times 15} & 0_{15\times 3} & 0_{15\times 3} & 0_{15\times 1}\\ 0_{3\times 3} & 0_{3\times 3} & 0_{3\times 3} & 0_{3\times 9} & 0_{3\times 15} & 0_{3\times 3} & 0_{3\times 3} & 0_{3\times 1}\\ 0_{3\times 3} & 0_{3\times 3} & 0_{3\times 3} & 0_{3\times 9} & 0_{3\times 15} & 0_{3\times 3} & 0_{3\times 3} & 0_{3\times 1}\\ 0_{1\times 3} & 0_{1\times 3} & 0_{1\times 3} & 0_{1\times 9} & 0_{1\times 15} & 0_{1\times 3} & 0_{1\times 3} & 0_{1\times 1} \end{array}\right]$$

The elements of the F are shown as follows: $F_{12}=\left[\begin{array}{ccc} 0 & 0 & \frac{1}{R_{N}\;+\;h}\\ 0 & 0 & \frac{\tan L}{R_{N}\;+\;h}\\ \frac{-1}{R_{M}\;+\;h} & 0 & 0 \end{array}\right]$, $F_{32}=\left[\begin{array}{ccc} \frac{1}{R_{M}\;+\;h} & 0 & 0\\ 0 & 0 & \frac{1}{\left(R_{M}\;+\;h\right)}\\ 0 & 1 & 0 \end{array}\right]$, $F_{13}=\left[\begin{array}{ccc} -\omega_{ie}\sin L & 0 & \frac{-v_{E}}{{\left(R_{N}\;+\;h\right)}^{2}}\\ \omega_{ie}\cos L\;+\;\frac{v_{E}}{\left(R_{N}\;+\;h\right)cos^{2}L} & 0 & \frac{-v_{E}\tan L}{{\left(R_{N}\;+\;h\right)}^{2}}\\ 0 & 0 & \frac{v_{N}}{{\left(R_{M}\;+\;h\right)}^{2}} \end{array}\right]$, $F_{14}=-C_{b}^{n}$$\left[\begin{array}{cccc} N_{x}^{g}I_{3\times 3} & \left[\begin{array}{c} 0_{1\times 2}\\ N_{y}^{g}I_{2\times 2} \end{array}\right] & \left[\begin{array}{c} 0_{2\times 1}\\ N_{z}^{g} \end{array}\right] & I_{3\times 3} \end{array}\right]$, $F_{22}=-\left[(2\omega_{ie}^{n}\;+\;\omega_{en}^{n})\times \right]\;+\;\left[v_{e}^{n}\times \right]F_{12}$, $F_{23}=\left[v_{e}^{n}\times \right]$$\left[F_{13}\;+\;\left[\begin{array}{ccc} -\omega_{ie}\sin L & 0 & 0\\ \omega_{ie}\cos L & 0 & 0\\ 0 & 0 & 0 \end{array}\right]\right]$, $F_{25}=C_{b}^{n}\left[\begin{array}{cccccc} N_{x}^{a}I_{3\times 3} & N_{y}^{a}I_{3\times 3} & N_{z}^{a}I_{3\times 3} & I_{3\times 3} & {\left(N^{a}\right)}^{2} & N^{g} \end{array}\right]$, and $F_{33}=\left[\begin{array}{ccc} 0 & 0 & \frac{-v_{N}}{{\left(R_{M}\;+\;h\right)}^{2}}\\ \frac{v_{E}\sin L}{\left(R_{N}\;+\;h\right)cos^{2}L} & 0 & \frac{-v_{E}}{{\left(R_{N}\;+\;h\right)}^{2}\cos L}\\ 0 & 0 & 0 \end{array}\right]$. Here, $N_{i}^{g}$ is the output of the three gyroscopes, and $N_{i}^{a}$ denotes the output of the three PIGAs. According to the analysis of the inner lever arm and the analysis of the time asynchronous error of the gyro accelerometer, the element matrices $F_{27}$ and $F_{26}$ in the state transition matrix can be obtained:(22)$$\begin{array}{c} F_{27}=M_{b}^{a}\cdot \left[{\left(\omega_{ib}^{b}\times \right)}^{2}+\left({\dot{\omega }}_{ib}^{b}\times \right)\right]\\ F_{28}=C_{b}^{n}\omega_{ib}^{b}\times f_{SF}^{b} \end{array}$$

The outer lever arm effect is mainly reflected in the observation equation in the Kalman filter. During the rotation of the indexing mechanism, the attitude error cannot be obtained in real time. However, during the rotation of the indexing mechanism, after compensating the outer lever arm, the observed speed and position are both 0.
(23)$$Z=\mathrm{HX}+V=\left[\begin{array}{c} v_{e}^{n}+C_{b}^{n}\left[\left(\omega_{ib}^{b}-C{{}_{b}^{n}}^{T}\omega_{ie}^{n}\right)\times \delta l^{b}\right]-v_{obv}\\ p+diag\left(\frac{1}{R{}_{M}+h},\frac{1}{(R_{N}+h)\cos L},1\right)C_{b}^{n}\delta l^{b}-p_{obv} \end{array}\right]$$

Therefore, the observation transition matrix H can be written as:(24)$$H=\left[\begin{array}{ccccccc} H_{11} & I_{3\times 3} & H_{13} & H_{14} & 0_{3\times 15} & C_{b}^{n}\left[\omega_{eb}^{b}\times \right] & 0_{3\times 7}\\ H_{21} & 0_{3\times 3} & H_{23} & 0_{3\times 9} & 0_{3\times 15} & H_{26} & 0_{3\times 7} \end{array}\right]$$

The elements of the H are shown as follows: $H_{11}=\left[\left(C_{b}^{n}\left(\omega_{eb}^{b}\times l^{b}\right)\right)\times \right]-C_{b}^{n}\left[l^{b}\times \right]C_{n}^{b}\left[\omega_{ie}^{n}\times \right]$, $H_{13}=C_{b}^{n}\left[l^{b}\times \right]C_{n}^{b}\left[\begin{array}{ccc} -\omega_{ie}\sin L & 0 & 0\\ \omega_{ie}\cos L & 0 & 0\\ 0 & 0 & 0 \end{array}\right]$, $H_{14}=-C_{b}^{n}\left[l^{b}\times \right]\left[\begin{array}{cccc} N_{x}^{g}I_{3\times 3} & \left[\begin{array}{c} 0_{1\times 2}\\ N_{y}^{g}I_{2\times 2} \end{array}\right] & \left[\begin{array}{c} 0_{2\times 1}\\ N_{z}^{g} \end{array}\right] & I_{3\times 3} \end{array}\right]$, $H_{21}=diag\left[\frac{1}{R{}_{M}\;+\;h},\frac{1}{(R_{N}\;+\;h)\cos L},1\right]\left[\left(C_{b}^{n}l^{b}\right)\times \right]$, $H_{23}=\left[\begin{array}{ccc} 1 & 0 & \frac{-l_{x}^{n}}{{\left(R_{M}\;+\;h\right)}^{2}}\\ \frac{l_{y}^{n}\sin L}{\left(R_{N}\;+\;h\right)\cos^{2}L} & 1 & \frac{-l_{y}^{n}}{{\left(R_{N}\;+\;h\right)}^{2}\cos L}\\ 0 & 0 & 1 \end{array}\right]$,$H_{26}=diag\left[\frac{1}{R{}_{M}\;+\;h},\frac{1}{(R_{N}\;+\;h)\cos L},1\right]$.

Since both the state equation and the observation equation are linear, we can use KF to estimate the error of the IMU.

Based on the previous analysis, the rotation path of the self-calibration utilizing angular velocity observation can be designed in Table 1:

The entire rotation path includes 18 rotation stages with a duration of 0.5 h; in the rotation stages 1–18, each rotation stage has a duration of 90 s, including the rotation movement and parking with an angular velocity of 20, where the purpose of rotation stages 1–10 is to excite and decouple all system-level self-calibration parameters; rotation stages 11–19 are mainly designed for rotation path design principle, whose purpose is to make all self-calibration parameters are fully estimated, especially the gyro bias error term. In addition, the initial alignment of the static base is performed for 120 s of coarse alignment and 180 s of fine alignment before self-calibration.

The self-calibration process is designed as follows:

As shown in Figure 4, the designed IMU consists of fiber optic gyroscopes (FOGs) and PIGAs, and the angular velocity observation is based on the FOG’s angular velocity output. Utilizing the error models of FOG and PIGA, the state equation and measurement equation of KF can be derived, substitute the two equations into KF’s time update and measurement update, the calibration results of the PIGA-based IMU can be obtained.

## 4. Experimental Results and Analysis

For the purpose of verifying the feasibility and effectiveness of the proposed online self-calibration method utilizing angular velocity observation for ultra high accuracy PIGA-based IMU. A self-calibration test is conducted to evaluate the accuracy of the calibration parameters. The circuit design of the embedded calculation and collect module is shown in Figure 5.

The flow chart of the inertial navigation signal is shown in Figure 5. The FOG transmits information such as uncalibrated angular increment, mechanical frequency jitter, and amplitude jitter to the Field Programmable Gate Array (FPGA) through the 3.3 V TTL level, and the current signal of the accelerometer passes through the I/F. After the module is converted into a frequency signal, it is transmitted to the FPGA. It should be noted that the FOG has a delay of 4 ms due to the low-pass filtering process. Therefore, the clock phase of the FPGA sampling signal to the DSP motion control module needs to be shifted to the left by 4 ms phase relative to the sampling signal to the FOG. In this way, the encoder angle information transmitted by the DSP motion control module to the FPGA through the serial port is synchronized with the output information of the IMU in time, and no related errors will be caused during the attitude demodulation process. After processing the relevant information, the FPGA transmits it to the DSP navigation module through EMIF, and also gives a 200 Hz square wave signal to the GPIO port of the DSP as the solution cycle (timed interrupt). Under the condition of large airborne dynamics, the operation rate of 200 Hz cannot meet the accuracy requirements. Therefore, in the process of the 4 k sampling of the IMU, the signal will not be accumulated, but will be latched and sent to the FPGA through EMIF.

During the process of calibration, the output of PIGAs and FOGs are shown in Figure 6.

It can be seen from Figure 6 that the PIGA’s output is related to the rotation process, which verifies the error model derived in Section II. The accuracy of FOG we utilize in this designed IMU is 0.002${}^{\circ }$/h (10 s, 1σ), with a 10 ppm (1σ) of scale factor repeatability.

The designed RINS is fixed in the marble, and the algorithm is implemented on a digital signal processor (DSP) chip. We use the method in ref. [16] as a comparison, and the self-calibration process lasts 30 min. In addition, we use a high-precision three-axis turntable to calibrate the IMU parameters as a reference. This method requires a high-precision turntable, and the IMU needs to be removed from the dual-axis RINS. For example, the accuracy of the dual-axis turntable is not high (especially horizontal accuracy). Traditional methods are described in [13]. The estimated curves of the IMU parameters are shown in Figure 7, Figure 8 and Figure 9:

Particularly, we draw the angular velocity sensitivity curve of PIGA separately in Figure 10 to perform a separate analysis of its convergence.

It can be seen from Figure 7 to Figure 9 that the errors added to the model do not affect the convergence of the IMU bias, scale factors, and installation angles. In Figure 10, The PIGA’s angular velocity coupling factors start to converge when the turntable rotates. To better discuss the experimental effect of this method, the estimated parameters are summarized in Table 2:

As shown in Table 2, the estimation accuracy of the proposed method is better than the traditional method, especially the calibration parameters of gyros. In addition, we find that the estimation results of PIGA’s angular velocity coupling factors are very close to the results utilizing high accuracy offline calibration method. The errors of gyro biases estimated by the traditional method are 0.012${}^{\circ }$/h to 0.023${}^{\circ }$/h, using the proposed method, the errors are only within 0.003${}^{\circ }$/h. The errors of the FOGs’ scale factors estimated by the traditional method are more than 15 ppm.

The self-calibration experiment results show that the propsoed method can not only estimate the PIGA’s angular velocity coupling factors, but also improve the gyroscope calibration parameters when utilzing the PIGA-based IMU.

## 5. Conclusions

Here, we propose an online self-calibration method utilizing angular velocity observation. Experimental results indicate that the proposed method accurately estimates the PIGA’s angular velocity coupling factors and improves the calibration accuracy by up to 0.02 m/s/pulse simultaneously, compared with the traditional self-calibration method (with an accuracy of 0.2 m/s/pulse) for PIGA-based IMU. After compensating for the PIGA’s angular velocity coupling factors, the navigation dynamic accuracy can be greatly improved. The self-calibration method also simplifies the calibration process and calibration implementation conditions, which make it possible to perform online-calibration without disassembling it and returning it to the factory for calibration.

There are still some error mechanisms that are understudied. In the future, research on decoupling the calibration of angular velocity and acceleration coupling coefficients will be carried out to improve the accuracy of PIGA continuously.

## Figures and Tables

**Figure 1 sensors-22-08136-f001:**
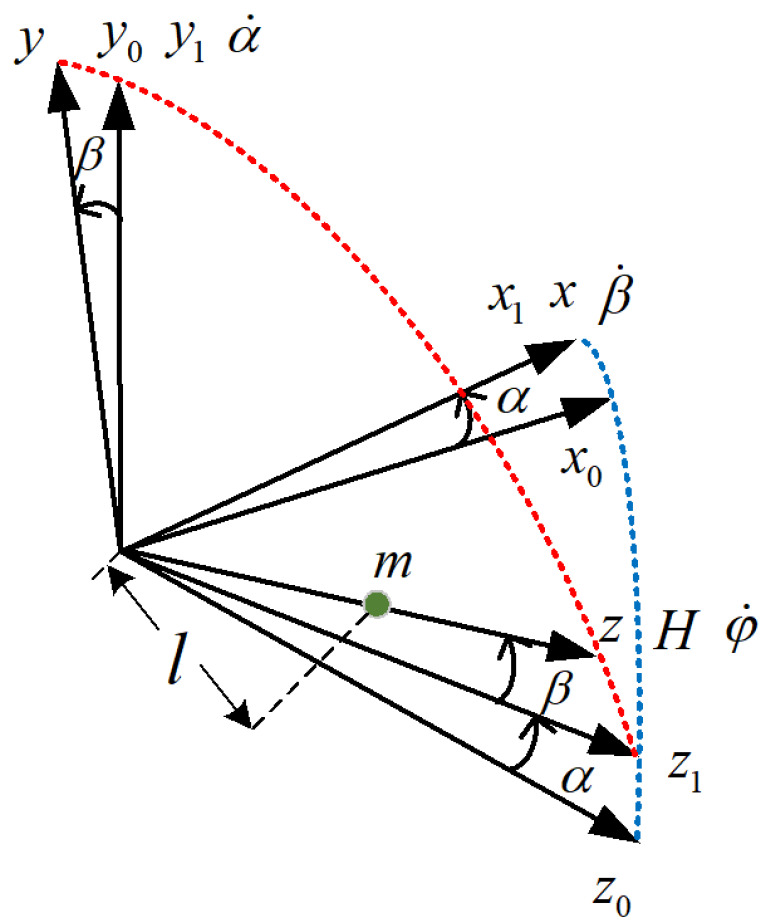
Internal coordinate system of PIGA.

**Figure 2 sensors-22-08136-f002:**
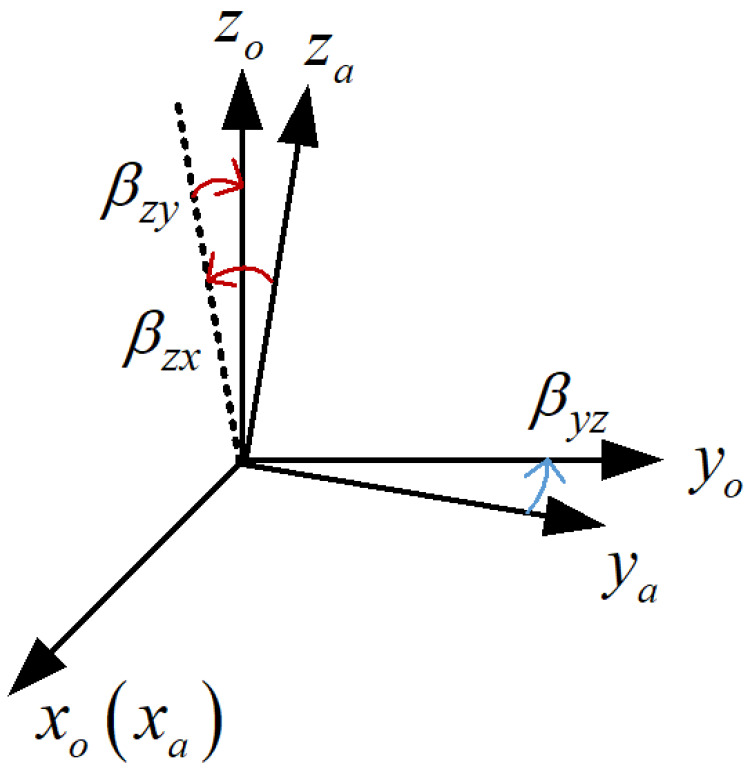
Relationship between accelerometer non-orthogonal coordinate frame and sensitive orthogonal coordinate frame.

**Figure 3 sensors-22-08136-f003:**
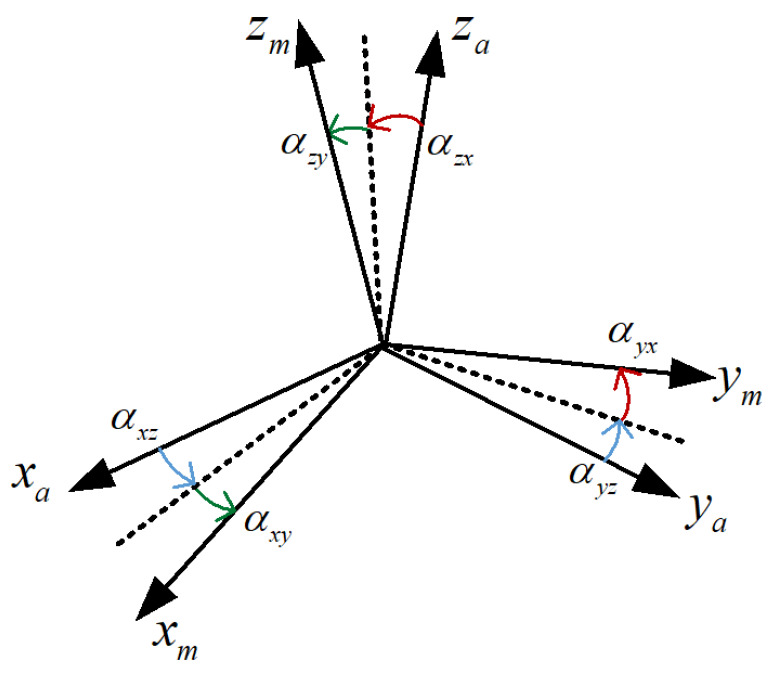
Accelerometer non-orthogonal coordinate frame.

**Figure 4 sensors-22-08136-f004:**
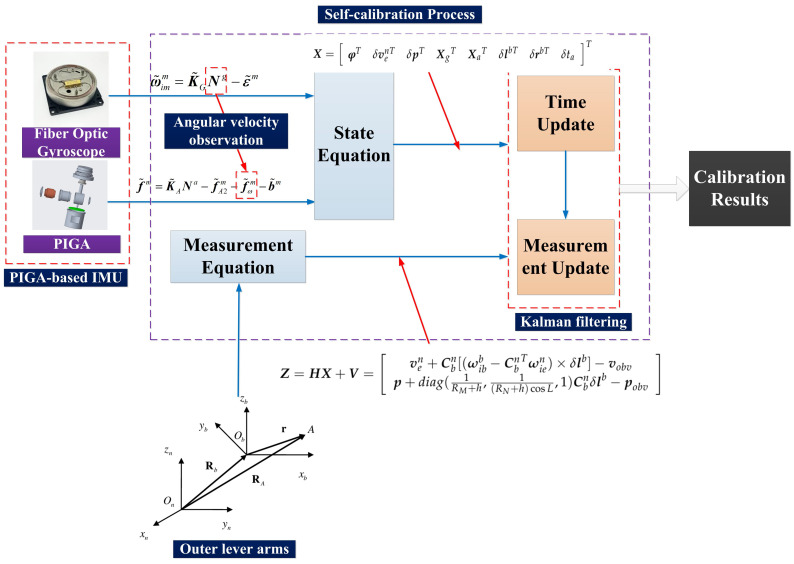
Self-calibration process utilizing angular velocity observation.

**Figure 5 sensors-22-08136-f005:**
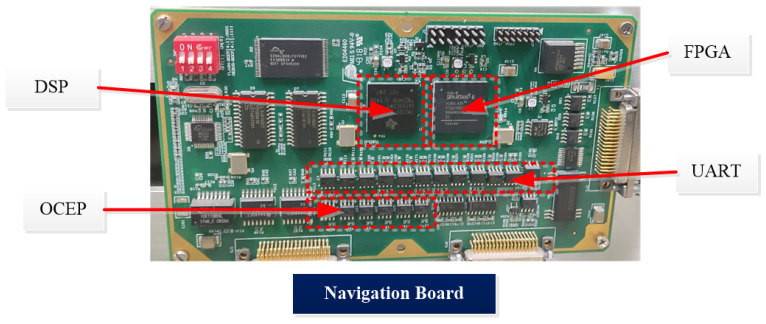
The circuit design of the embedded calculation and collect module.

**Figure 6 sensors-22-08136-f006:**
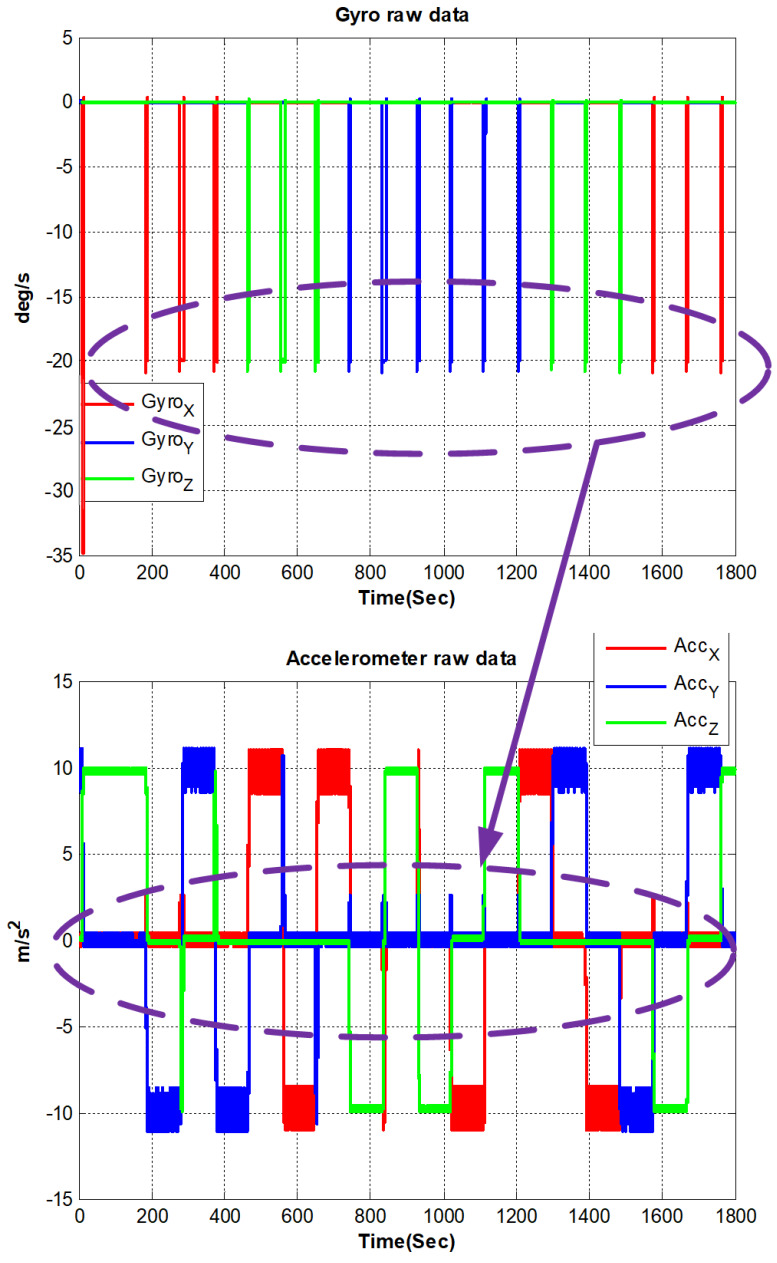
Raw data of PIGA-based IMU during self-calibration process.

**Figure 7 sensors-22-08136-f007:**
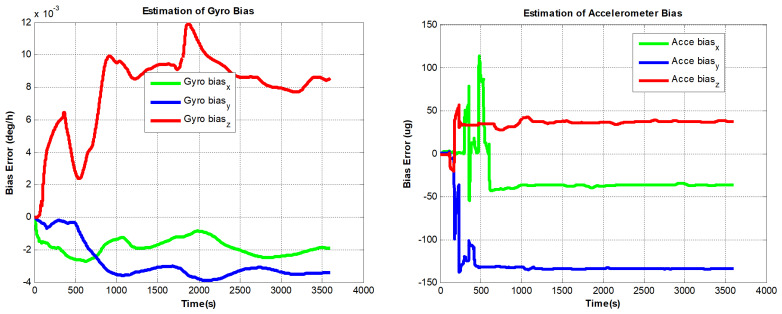
PIGA-based IMU biases estimation curves.

**Figure 8 sensors-22-08136-f008:**
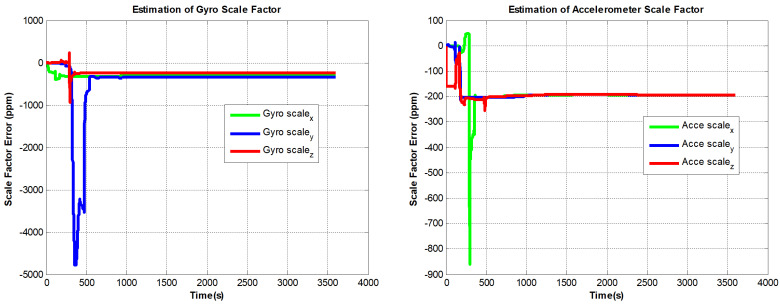
PIGA-based IMU scale factors estimation curves.

**Figure 9 sensors-22-08136-f009:**
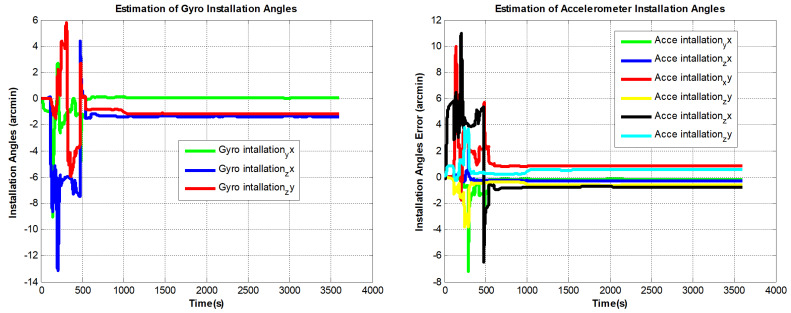
PIGA-based IMU installation angles estimation curves.

**Figure 10 sensors-22-08136-f010:**
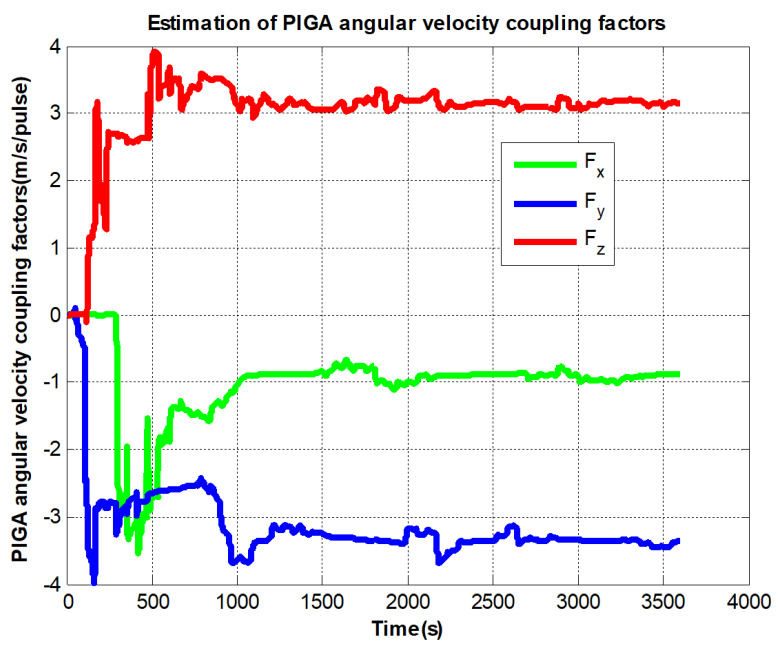
PIGA’s angular velocity coupling factors estimation curves.

**Table 1 sensors-22-08136-t001:** Rotation path of self-calibration process.

Time	Rotation Axis (Inner (I) (*z*-Axis of IMU)/Outer (O) (*x*-Axis of IMU))	Rotation Angle along I/O Axis	Attitude after Rotation (X)	Attitude after Rotation (Y)	Attitude after Rotation (Z)
0 s	-	-	East	North	Upward
180 s	O	+90${}^{\circ }$	East	Upward	South
270 s	O	+180${}^{\circ }$	East	Downward	North
360 s	O	+180${}^{\circ }$	East	Upward	South
450 s	I	+90${}^{\circ }$	Upward	West	South
540 s	I	+180${}^{\circ }$	Downward	Upward	South
630 s	I	+180${}^{\circ }$	Upward	West	South
720 s	O	+90${}^{\circ }$	South	West	Downward
810 s	O	+180${}^{\circ }$	North	West	Upward
900 s	O	+180${}^{\circ }$	South	West	Downward
990 s	O	+90${}^{\circ }$	Downward	West	North
1080 s	O	+90${}^{\circ }$	North	West	Upward
1170 s	O	+90${}^{\circ }$	Upward	West	South
1260 s	I	+90${}^{\circ }$	West	Downward	South
1350 s	I	+90${}^{\circ }$	Downward	East	South
1440 s	I	+90${}^{\circ }$	East	Upward	South
1530 s	O	+90${}^{\circ }$	East	South	Downward
1620 s	O	+90${}^{\circ }$	East	Downward	North
1710 s	O	+90${}^{\circ }$	East	North	Upward

**Table 2 sensors-22-08136-t002:** Estimation results of different methods.

Estimated Parameters	Proposed Method	Traditional Method	Reference Values
$\varepsilon_{x}$	−0.0189${}^{\circ }$/h	−0.0092${}^{\circ }$/h	−0.0176${}^{\circ }$/h
$\varepsilon_{y}$	0.0312${}^{\circ }$/h	0.0381${}^{\circ }$/h	0.0309${}^{\circ }$/h
$\varepsilon_{z}$	0.0852${}^{\circ }$/h	0.0786${}^{\circ }$/h	0.0843${}^{\circ }$/h
$\nabla_{x}$	423.23 μg	413.77 μg	423.71 μg
$\nabla_{y}$	−808.63 μg	−810.65 μg	−808.47 μg
$\nabla_{z}$	687.76 μg	692.65 μg	687.36 μg
$\delta k_{xx}^{g}$	100,063.42${}^{\circ }$/h/pulse	100,069.66${}^{\circ }$/h/pulse	100,063.76${}^{\circ }$/h/pulse
$\delta k_{yy}^{g}$	100,067.43${}^{\circ }$/h/pulse	100,068.78${}^{\circ }$/h/pulse	100,067.90${}^{\circ }$/h/pulse
$\delta k_{zz}^{g}$	100,053.78${}^{\circ }$/h/pulse	100,054.74${}^{\circ }$/h/pulse	100,053.12${}^{\circ }$/h/pulse
$\delta k_{xx}^{a}$	98,012.98 m/s${}^{2}$/pulse	98,013.12 m/s${}^{2}$/pulse	98,012.34 m/s${}^{2}$/pulse
$\delta k_{yy}^{a}$	98,015.76 m/s${}^{2}$/pulse	98,016.31 m/s${}^{2}$/pulse	98,015.48 m/s${}^{2}$/pulse
$\delta k_{zz}^{a}$	98,063.94 m/s${}^{2}$/pulse	98,062.52 m/s${}^{2}$/pulse	98,063.32 m/s${}^{2}$/pulse
$\delta k_{yx}^{g}$	3.487${}^{\prime }$	4.521${}^{\prime }$	3.654${}^{\prime }$
$\delta k_{zx}^{g}$	−2.653${}^{\prime }$	−3.987${}^{\prime }$	−2.76 ${}^{\prime }$
$\delta k_{zy}^{g}$	11.676${}^{\prime }$	10.455${}^{\prime }$	11.149${}^{\prime }$
$\delta k_{xy}^{a}$	9.421${}^{\prime }$	9.912${}^{\prime }$	9.122${}^{\prime }$
$\delta k_{xz}^{a}$	7.645${}^{\prime }$	6.938${}^{\prime }$	7.476${}^{\prime }$
$\delta k_{yx}^{a}$	1.567${}^{\prime }$	1.765${}^{\prime }$	1.543${}^{\prime }$
$\delta k_{yz}^{a}$	−5.141${}^{\prime }$	−5.267${}^{\prime }$	−5.134${}^{\prime }$
$\delta k_{zx}^{a}$	3.112${}^{\prime }$	3.983${}^{\prime }$	3.145${}^{\prime }$
$\delta k_{zy}^{a}$	6.653${}^{\prime }$	5.769${}^{\prime }$	6.790${}^{\prime }$
$\delta K_{w}$	−0.978 m/s/pulse	−1.176 m/s/pulse	−0.981 m/s/pulse
$\delta K_{w}$	−3.313 m/s/pulse	−3.026 m/s/pulse	−3.301 m/s/pulse
$\delta K_{w}$	3.121 m/s/pulse	3.389 m/s/pulse	3.112 m/s/pulse

## Data Availability

Not applicable.

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
