# Peer review of "An Improved Online Self-Calibration Method Utilizing Angular Velocity Observation for Ultra High Accuracy PIGA-Based IMU"

_sensors, 2022, doi:10.3390/s22218136_

Round 1

Reviewer 1 Report

This paper is very well written and contributes a simplified error model for online self-calibration model, which enables an improved online self-calibration method utilizing angular velocity observation. And the proposed scheme outperforms the state of the arts.

But there are some problems, which must be solved before it considered for publication.

There are some language and format problems in this paper. Some sentences in the paper contain grammatical and temporal errors, such as, line 67. And please check the format, e.g. Kyy in line 154, y should be in subscript, please check all. And there are some writing mistakes, e.g. the ‘The’ in line 158 and line 162 and the ‘High’ in line 301, which makes the article hard to understand.

In page 1, the ABSTRACT ,more quantitative metrics may be used in place of ‘effectively’. And, the conclusions needs more in it, as it’s more of an afterthought. The authors are suggested to highlight important findings and include afterthought of this work.

Author Response

We would like to thank the Chief Editor, the Associate Editor and the reviewers for their objective and thorough review of our paper. We have addressed all the reviewers’ comments in the following point-by-point response and changed the manuscript accordingly. Changes made in the revised manuscript are with a different text color.

Reviewer 2 Report

The examined paper is very interesting and high-quality. The article’s content is relevant to the scientific area of the Sensors Journal.

The article’s title represents the content and purpose of the article. The abstract is concise and relevant. The keywords are adequate. The Introduction and Abstract clearly identify the need and relevance of this research. The article structure is clear.

It contains explicitly required sections such as Introduction with references review, Modeling and Analysis, Method, Experimental results and analysis, and Conclusion. The improved online self-calibration method is sound and the results are useful. The paper contributes to the body of knowledge. The paper is technically sound. The provided references are applicable and sufficient.

There are some comments:

1) No numerical results of the study are displayed in the Abstract and Conclusions. The future of this study should be added to the Conclusions.

2) In Figure 4, the formulas are very small and hard to be discerned. It is also hard to distinguish inscriptions on the coordinate system at the bottom of Figure 4.

3) Minor remarks:

- there is a broken sentence in the first paragraph of section 2 at the bottom of page 2 (lines 78-79);

- in line 262 at the end of the sentence, please, put a period, not a colon;

- the phrase "as shown in Figure 6" should be added in lines 291-292;

- the phrase "as shown in Figures 7-9" should be added in lines 302-303;

- the phrase "(Figure 10)" should be added in lines 304-305.

In general, the article is very good. There are no remarks about its main content. Minor corrections are needed.

By considering the above, I recommend this paper for publication in the Sensors Journal.

Author Response

(The authors gave the same response as above.)

Round 2

Reviewer 1 Report

It’s glad to see the concerns that we proposed are properly addressed. However, there are some new problems appeared.

The chart format and order may need to be improved. For example, Table II is not clear and some information is obscured. And the repetition of the abstract and conclusions is slightly more. As CONCLUSIONS is more of an afterthought, we don’t need to say why we did this research, again. As for the repetitive part in the abstract and conclusion, I hope the author can express it in another way rather than copy it directly.

These problems should be solved before this paper considered for publication.

Author Response

(The authors gave the same response as above.)
